# Capacitive NO_2_ Detection Using CVD Graphene-Based Device

**DOI:** 10.3390/nano13020243

**Published:** 2023-01-05

**Authors:** Wonbin Ju, Sungbae Lee

**Affiliations:** 1Department of Physics and Photon Science, Gwangju Institute of Science and Technology, Gwangju 61005, Republic of Korea; 2Korea Institute of Energy Technology, KENTECH College, Naju 58330, Jeonnam, Republic of Korea

**Keywords:** graphene, nitrogen dioxide sensing, capacitive sensing, quantum capacitance, graphene field-effect transistor, nitrogen dioxide adsorbed graphene

## Abstract

A graphene-based capacitive NO_2_ sensing device was developed by utilizing the quantum capacitance effect. We have used a graphene field-effect transistor (G-FET) device whose geometrical capacitance is enhanced by incorporating an aluminum back-gate electrode with a naturally oxidized aluminum surface as an insulating layer. When the graphene, the top-side of the device, is exposed to NO_2_, the quantum capacitance of graphene and, thus, the measured capacitance of the device, changed in accordance with NO_2_ concentrations ranging from 1–100 parts per million (ppm). The operational principle of the proposed system is also explained with the changes in gate voltage-dependent capacitance of the G-FET exposed to various concentrations of NO_2_. Further analyses regarding carrier density changes and potential variances under various concentrations of NO_2_ are also presented to strengthen the argument. The results demonstrate the feasibility of capacitive NO_2_ sensing using graphene and the operational principle of capacitive NO_2_ sensing.

## 1. Introduction

Since the discovery of graphene, a two-dimensional carbon atomic layer prepared by mechanical exfoliation of highly oriented pyrolytic graphite, it has garnered the attention of many researchers due to its unique electronic, optical, and mechanical properties [1,2,3,4,5,6,7]. One of the promising applications is its usage as a sensing material for chemicals, owing to its electrical properties, large surface-to-volume ratio, and chemical stability [8]. The sensing ability of graphene extends to detection of a wide range of chemicals, such as NO_2_, CO, SO_2_, H_2_S, etc. [8,9,10,11]. Furthermore, functionalized graphene, defective graphene, and doped graphene have been studied to improve the selectivity and sensitivity of graphene-based sensing devices [12,13,14,15,16]. Graphene-based gas sensing devices have great of potential for detecting dangerous gas species [17], such as nitrogen dioxide (NO_2_), one of the dangerous air pollutants that can affect human health even in low concentrations [18,19]. In this respect, graphene-based NO_2_ sensing applications have been extensively studied for decades. However, most of related research has been focused on measuring a change in resistance (or conductance) when chemicals are adsorbed on a graphene surface [20,21,22,23].

Although graphene-based capacitive sensing applications have been studied in recent years with various chemicals and gas/vapors [24,25,26,27], an extensive study on capacitive NO_2_ sensing has not yet been reported. The total capacitance (*C*_total_) of a graphene-based capacitive sensing device consists of the series connection of geometrical capacitance (*C*_geo_) and quantum capacitance (*C*_Q_). The quantum capacitance effect is the main reason behind the changes in *C*_total_ upon the adsorption of chemical molecules on the graphene surface. However, due to the electrical characteristics of a system of capacitors, the system requires greater *C*_geo_ compared to the minimum *C*_Q_ (or at least equal to *C*_Q_), in order for *C*_total_ to be dominantly determined by *C*_Q_. This indicates that the sensitivity of the capacitive sensing device can be improved by enhancing the *C*_geo_ [24,28,29]. Further analyses of quantum capacitance provide electronic properties of materials, such as the density of states, carrier density, and potential variance [28,30,31,32,33], which allows us to confirm the working principle of the proposed device.

In this work, we report capacitive NO_2_ sensing performance of an Al back-gated G-FET. Various gas molecules (NH_3_, H_2_O, CO, O_2_, NO_2_, etc.) can be adsorbed on graphene surfaces due to various reasons on graphene surfaces, resulting in changes in electronic response. The issue of functionalizing a graphene surface to increase the selectivity of a target molecule is a completely different issue yet to be resolved. However, we can still test the electrical responses of graphene-based devices by creating isolated environments to test the sensing procedure. Furthermore, the adsorbed gas molecules on graphene can be easily detached by thermal treatment or UV exposure, which allows us to recycle the prepared devices for the test under various conditions. To investigate the capacitive response of graphene for NO_2_, we restricted the test gases to a NO_2_/N_2_ mixture and adopted UV exposure for the recovery process. Water molecules in the air affect the quantum capacitance of graphene by adsorbing it onto graphene surfaces [25]. Oxygen molecules can also affect the quantum capacitance of graphene by adsorbing onto graphene surfaces. However, diluted NO_2_ in dry air was detected well using graphene-based devices by measuring its resistance [34]. In order to use the proposed device in daily life, response tests for NO_2_ diluted in ambient air as a function of humidity should be further studied.

We focus on capacitive NO_2_ sensing performance, and later, the electronic properties of graphene exposed to NO_2_ molecules will be discussed to strengthen our argument regarding the working principle of the device. Changes in the capacitance of the device were measured before and after the vacuum test chamber was filled with various concentrations of NO_2_ (1–100 ppm balanced with pure nitrogen gas). Furthermore, the electronic properties, such as residual carrier density and potential variance at different NO_2_ concentrations, were extracted from the gate voltage-dependent *C*_total_. Our results demonstrate the feasibility of capacitive NO_2_ sensing and, thus, provide preliminary research for capacitive gas sensing using graphene.

## 2. Methods

### 2.1. Device Fabrication

A capacitive NO_2_ sensing device was fabricated using CVD-grown graphene on Cu foil (LG electronics). Monolayer graphene was confirmed by Raman spectroscopy, as shown in Appendix A. At first, 15 nm of Al back-gate electrode was deposited on a sapphire substrate using photolithography and electron beam evaporation. Soon after the sample was removed from the evaporation chamber, oxidation took place, forming a few nanometers-thick layer of oxidized Al (AlO*_x_*) on the surface of the Al gate electrode. Later, this extra thin layer of AlO*_x_* would serve as a high-*k* insulating layer for the graphene-based capacitor with a relatively large geometric capacitance. Next, CVD graphene was transferred onto the sapphire substrate. The Cu foil with top-side graphene was spin-coated with poly(methyl methacrylate) (PMMA) (4% in anisole) and then baked on a hot plate at 120 °C for 10 min. The reverse side of the graphene (lacking PMMA) was etched using O_2_ reactive-ion etching (RIE). Afterwards, PMMA/graphene on the Cu foil was placed in ammonium persulfate solution (5 wt% in distilled water) for more than 3 h to remove the Cu foil. After finishing the Cu foil etching, the graphene/PMMA film was transferred to distilled water three times to rinse the residual chemicals beneath the graphene. Subsequently, the PMMA-supported graphene film was transferred to the target substrate. Then, the sample was dried in an oven at 60 °C for more than 5 min. The PMMA layer on the substrate was removed by soaking in acetone for more than an hour. The graphene channel was patterned using photolithography and O_2_ RIE. Finally, 40 nm of Au source and drain electrodes were formed on the graphene channel using photolithography and electron beam evaporation. A cross-sectional view and an optical microscopy image of the fabricated device are shown in Figure 1a,b, respectively. The active capacitive sensing area is graphene on the Al back-gate electrode, as marked with the red dashed line in Figure 1a.

### 2.2. Measurement Setup

Gas sensing measurements were carried out in a test chamber equipped with pure nitrogen (N_2_ 99.999%) and 100 ppm NO_2_ (balanced with N_2_) gas, mass flow controllers (MFCs), a UV lamp (TUV 4W G4T5, Philips, Seoul, Republic of Korea), a measuring instrument (4200-SCS, Keithley, Cleveland, OH, USA), a mechanical pump, and a quartz window, as shown in Figure 2. To measure the NO_2_ sensing performance, various concentrations (1–100 ppm) of NO_2_ gas are introduced into the test chamber by varying the ratio of the flow rate of 100 ppm NO_2_ and pure N_2_ using MFCs at a total flow rate of 10 standard liters per minute (SLM). The gas sensing performance of the device under test (DUT) is monitored by the change in capacitance caused by gas molecule adsorption on the graphene surface. The UV lamp outside the chamber is used during the recovery process; the distance between the device under test and the UV lamp is approximately 7 cm. All electrical measurements were carried out using a Keithley 4200-SCS at room temperature.

### 2.3. Measurement Flow

Before beginning the sensing measurements, the mechanical pump evacuates the chamber to its initialized state at a pressure of around 10^−3^ torr (the black line in Figure 3). While measuring capacitance in real-time under vacuum (stage ➀), the gas mixture of NO_2_/N_2_ with a target concentration is injected into the initialized test chamber until the pressure reaches atmospheric pressure (~760 torr). Then, all the valves are closed to isolate the chamber. The device is exposed to an NO_2_ environment for approximately 2 h to check capacitance saturation time (stage ➁). To characterize sensing performance, the response is calculated using the following equation:(1)$$\mathrm{Response}=\left|\frac{C_{g}-C_{0}}{C_{0}}\right|\times 100\%$$
where *C*_g_ and *C*_0_ are the capacitance before and after the exposure to NO_2_, respectively [35]. Then, *C*_total_ is measured by sweeping the back-gate voltage, *V*_BG_ from +2 V to −2 V, to analyze the electronic properties of graphene exposed to different concentrations of NO_2_ (stage ➂).

Thermal treatment and UV illumination can be used to detach adsorbates on graphene [8,22,34,36]. The UV illumination method was adopted for our recovery process because of its timely response compared to thermal treatment in our measurement setup. The UV light generates electron–hole pairs that detach the NO_2_ adsorbates via hole recombination, e.g., hole + NO_2_^−^ → NO_2_ (gas), on graphene [22,37]. Though appropriate UV exposure in ambient air improves the recovery process, overexposure to UV light under ambient condition may generate ozone, causing damage to the carbon–carbon bonding in graphene and creating defects. However, UV irradiation on graphene in inert gas or vacuum does not significantly affect the defect sites [37,38]. Furthermore, the graphene sensors for NO_2_ have been known to exhibit excellent durability and reliability for the UV-assisted recovery process [22].

Thus, during the recovery process, the chamber is first evacuated by a mechanical pump to reduce the residual NO_2_ molecules and possible introduction of O_2_ in the chamber (stage ➃). While pumping, the UV light is turned on to detach the adsorbed gas molecules from the graphene surface (stage ➄) and then tuned off (stage ➅) until the capacitance of the device returns to the initial state (the capacitance before NO_2_ exposure). Then, the response, gate sweeping, and recovery processes are repeated with different concentrations of NO_2_.

## 3. Results and Discussion

To evaluate the NO_2_ sensing performance, the real-time capacitive response of the device at different NO_2_ concentrations was investigated, as shown in Figure 4a. This is the collection of capacitance response data acquired during the early period of stage ➁ under various NO_2_ concentrations. Note that actual measured capacitance for calculating the response in Figure 4a is shown in Appendix A. As illustrated in previous section, the test chamber starts to be filled with NO_2_ with a desired concentration from 1 to 100 ppm while measuring the capacitance in real-time. The *t* = 0 value in Figure 4a is set to the beginning of the gas injection. The initial value of capacitance (before the beginning of gas injection) for all trials was set to an almost constant value for a fair comparison. The overall capacitive response increases proportionally to the concentration of NO_2_ after the test chamber is filled with the test gas. The NO_2_ molecules in the test chamber adhere to the graphene surface and act as electron acceptors; they attracts electrons from graphene, leaving holes [17]. The hole density causes the increased *C*_Q_ of the graphene exposed to NO_2_, because *C*_Q_ of graphene is proportional to the square root of the carrier density of graphene [28]. The measured capacitance is the series connection of *C*_geo_ of AlO*_x_* and *C*_Q_ of graphene. This leads to changes in the measured capacitance of the device. The capacitive response immediately increases after the relatively high concentrations of 10–100 ppm NO_2_ are injected into the test chamber. On the other hand, the response to the relatively low concentrations of 1–5 ppm NO_2_ begins to change in a few seconds. Relatively high concentrations of NO_2_ are enough to dope graphene at the moment of the injection, so the response graph shows a dramatic change in value at the very early stages of exposure. The response at low concentrations of NO_2_ takes some time to dope graphene since, NO_2_ molecules in diluted gases are not enough to dope graphene at the moment of the injection. However, at the lower concentrations, our device can detect the differences in concentrations with higher precision within a moderately allowed time interval.

The response values at *t* = 75 s (roughly when the chamber pressure reaches atmospheric pressure) and 150 s are plotted as a function of NO_2_ concentration, as shown in Figure 4b. The response at 75 s proportionally increases with increasing NO_2_ concentration in the two regions, namely 1–5 and 10–100 ppm. Although the response time under the higher NO_2_ concentration mixture is much faster simply due to the introduction of the large number of NO_2_ molecules in the system, the slope of response graph for lower concentrations (1–5 ppm) at the early stages of exposure is greater compared to that of larger concentration cases (10–100 ppm), which means the change in response provides a more precise distinction between various NO_2_ concentrations below 10 ppm. We believe that this distinction is drawn simply due to the total number of NO_2_ molecules our device can hold, and that the numbers can be modulated further by redesigning the size of the active channel. The change in the slope shown in Figure 4b can be explained by a shift in *C*_total_ as a function of the *V*_G_ curves. A detailed discussion will be given later. The response at 150 s (solid symbols) is increased with a similar trend to the response at 75 s. The changes between 75 and 150 s at high NO_2_ concentrations are lower than the low concentrations, which means that the response at high concentrations saturates faster than at low concentrations. To make sure of the effect of the pure N_2_ as a diluent gas, the response of 0 ppm NO_2_ was tested. When the test chamber was filled with pure N_2_, the response was changed by lower than 0.1% for 200 s. Impurities (the rest of 99.999% N_2_) in pure N_2_ do not significantly affect the capacitive sensing for 200 s, confirming that the N_2_ is only used to dilute 100 ppm NO_2_.

Figure 5a shows the different sensing behavior of the differential capacitance (*dC*/*dt*) as a function of time. Once the graphene is exposed to 10–100 ppm NO_2_, *dC*/*dt* starts to immediately increase until it reaches the maximum, and then it approaches 0; on the other hand, *dC*/*dt* exposed to 1–5 ppm NO_2_ shows a relatively slowly increase to the maximum and then approaches 0. The *dC*/*dt* at 6.6 s (at the maximum value of *dC*/*dt* at 100 ppm NO_2_) is shown in Figure 5b. The *dC*/*dt* is a nearly linear relation with the NO_2_ concentration, suggesting that the device can detect a high concentration of NO_2_ by measuring *dC*/*dt* at the early stage. Figure 5c shows *dC*/*dt* at the moment the pressure of the chamber reaches atmospheric pressure (75 s). The *dC*/*dt* has an inversely linear relation with NO_2_ concentrations, except for 0 ppm, which is a different result from Figure 5b. These indicate that the capacitive response at the high NO_2_ concentration saturates faster than at the low NO_2_ concentration, as expected.

Response change for an up to 2 h time span is shown in Figure 6a. Note that the response curves in Figure 4a depict the early part of the response change in Figure 6a (up to 200 s). After some time, the responses at all concentrations seemingly exhibit no significant change. It is probably the best time to address the fact that the capacitive response presented in this research shows faster responses compared to the current response given in Appendix A. Even though the response in current measurements seems a lot higher in values when fully saturated, the capacitive measurements show sharper changes at the initial response, one of the most essential features that the industry requires when it comes to detecting hazardous molecules at the earliest possible stage. This becomes clearer when the gate voltage-dependent capacitance in Figure 6b is compared to the gate voltage-dependent drain current presented in Appendix A near the Dirac point. The curvature means how sharply a curve bends at a given point; the higher the curvature, the more bent the curve is. The curvature of the gate-dependent current and capacitance shows a peak at the Dirac point voltage as shown in Appendix A, respectively. The full width at half maximum (FWHM) of the capacitance curvature is smaller than the FWHM of the current curvature. The gate-dependent capacitance is more sharply curved in a narrower range near the Dirac point, indicating the greater detection possibility at the very early stages of NO_2_ introductions into the system.

The response at 0 ppm NO_2_ (99.999% N_2_) is almost constant for approximately 15 min after the test chamber is filled. The response starts to slowly increase after a quarter hour. We speculate that 99.999% N_2_ as a diluent containing 0.001% unexpected impurities acting as acceptors slowly dopes graphene. However, when 100 ppm NO_2_ is diluted by 99.999% N_2_, responses of 1–50 ppm NO_2_ (even with smaller concentrations) do not increase after a quarter hour, indicating that NO_2_ is binding with graphene far more strongly compared to any other possible impurities that might be present in the balancing gas (N_2_). This also suggests that N_2_ is a proper diluent for adjusting NO_2_ concentration in this test.

Figure 6b shows the gate voltage-dependent *C*_total_ from 2 V to −2 V after finishing the NO_2_ sensing measurement (in 2 h), in order to extract carrier density and change in potential fluctuations induced by adsorbed NO_2_ molecules [29,33,39]. The absorbed NO_2_ molecules shift the Dirac point voltage (*V*_DP_) to the positive gate voltage side and causes a rounding of the curves near the *V*_DP_, which is the voltage with the minimum *C*_total_. The shift in *V*_DP_ can reasonably be explained by charge transfer between the graphene and NO_2_ molecules [36]. Adsorbed NO_2_ molecules act as electron acceptors, which perform a similar function as the applied negative gate voltage [40].

The upward shift of the *C*_total_ curves is attributed to the enhancement of local geometrical capacitance caused by NO_2_ molecules near graphene. The NO_2_ molecules which are polar molecules can be intercalated between graphene and the Al electrode, similar to intercalated H_2_O molecules between a substrate and graphene [25]. The intercalated NO_2_ molecules enhance the effective dielectric constant of the insulating layer in the graphene/AlO*_x_*/Al capacitor. The NO_2_ molecules on graphene are also one of possible contributions to the enhanced electric field originating from the gate electrode. Due to the low density of the state of graphene, the graphene does not screen all of the electric field, which affects NO_2_ molecules on graphene when *C*_geo_ is comparable to the minimum *C*_Q_. The NO_2_ molecules on the graphene are aligned by the penetrated electric field, resulting in the enhancement of the effective geometrical capacitance.

The *C*_total_ curves in Figure 6b support the response curve in Figure 4a (also Figure 6a). As NO_2_ concentration in the chamber increases, the capacitance curves shift up to the right, resulting in changes in *C*_total_ at *V*_BG_ = 0 V. This is the mechanism of capacitive NO_2_ sensing. The dramatic change in the *C*_total_ curve at 0 ppm near *V*_DP_ explains the slope in the two regions in Figure 4b. Furthermore, Δ*R*_1ppm_ (the difference between the response with 1 ppm and the response with 0 ppm) after 2 h exposure is 23.8%, suggesting that the device can detect sub-ppm NO_2_ through the capacitance measurement. The expected response at concentrations between 0 ppm and 1 ppm would be between 11.8% and 35.6%.

To investigate the electronic properties of NO_2_ adsorbed graphene, the gate voltage-dependent *C*_total_ in Figure 6b was fitted using the microscopic model of quantum capacitance in graphene suggested by Xu et al. [39], with *C*_geo_, parasitic capacitance, and δ*V* (potential variance) as fitting parameters [29]. The extracted δ*V* of graphene exposed to NO_2_ for 2 h as a function of NO_2_ concertation is plotted in Figure 6c. Charge transfer between graphene and NO_2_ molecules causes a local potential fluctuation near the adsorbed NO_2_ molecules. Since the local potential fluctuations are affected by charged impurities near graphene, such as the density of adsorbed NO_2_ molecules, δ*V* becomes higher as NO_2_ concentration increases. In addition, the residual carrier densities were calculated from the quantum capacitance minimum extracted from fitting using the equation $C_{Q}=2e^{2}\sqrt{n}/\left(\hbar v_{F}\sqrt{\pi }\right)$. The residual carrier density has a similar trend to the potential variation. The results confirm that adsorbed NO_2_ molecules cause charge transfer from graphene and produce potential fluctuation. Holes produced from the charge transfer shift the *V*_DP_ to the right. The potential fluctuations round the capacitance curve near *V*_DP_. These changes are determined by absorbed NO_2_ molecules and explain changes in the capacitance at zero gate voltage in Figure 4a (also Figure 6a).

## 4. Conclusions

In this study, Al back-gated G-FET was fabricated to measure capacitive NO_2_ sensing performance and the electronic properties of NO_2_ adsorbed graphene. The quantum capacitance effect caused by enhanced *C*_geo_ of naturally oxidized Al allowed for capacitive sensing. The capacitance of the device exposed to 1–100 ppm NO_2_ was changed by 21–51% compared to the initial capacitance at *t* = 150 s. The Δ*R*_1ppm_ at *t* = 150 s is 21%, indicating that the device would detect sub-ppm NO_2_ by the capacitance measurement of the device. Furthermore, the capacitive NO_2_ sensing mechanism is explained by the gate voltage-dependent *C*_total_. Adsorbed NO_2_ molecules on the graphene surface shift *V*_DP_ as a function of concentrations of NO_2_. Carrier density and potential variations in the device caused by absorbed NO_2_ molecules were extracted from quantum capacitance by fitting. These results demonstrate the fundamental understanding of the absorbed NO_2_ effect on graphene from capacitance measurement, as well as that capacitive NO_2_ sensing is possible by enhancing the *C*_geo_ of graphene-based devices.

## Figures and Tables

**Figure 1 nanomaterials-13-00243-f001:**
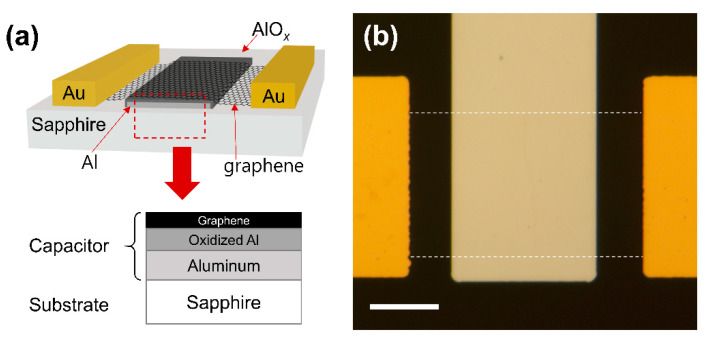
(**a**) Cross-sectional view of the device. The red dashed box represents the active capacitor area. (**b**) Optical microscope image of the fabricated device. The graphene channel is marked with white dashed lines. The scale bar is 15 μm.

**Figure 2 nanomaterials-13-00243-f002:**
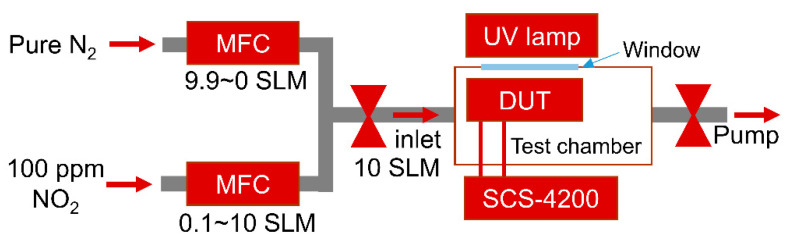
Schematic diagrams of the gas sensing measurement setup.

**Figure 3 nanomaterials-13-00243-f003:**
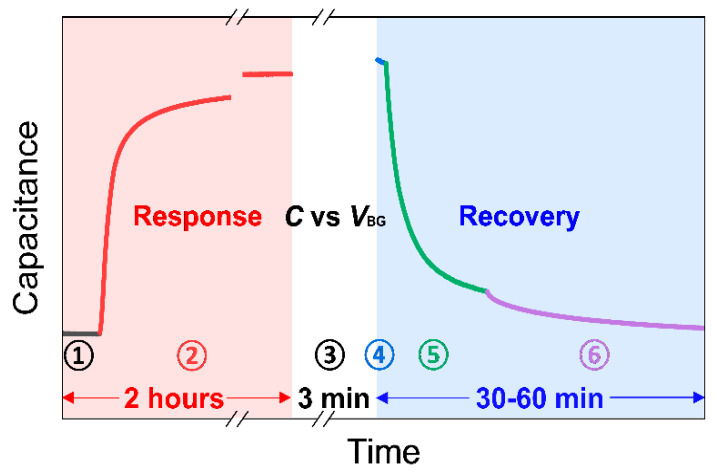
An example of a single measurement process consisting of ‘Response’ period, ‘*C*_total_ vs. *V*_BG_’ period, and ‘Recovery’ period. Detailed explanations for each stage number from ➀ to ➅ are given in Section 2.3.

**Figure 4 nanomaterials-13-00243-f004:**
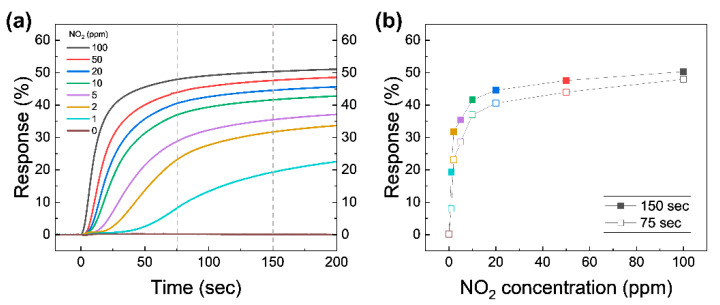
(**a**) Capacitive response as a function of time for various NO_2_ concentrations. Atmospheric pressure is reached in 75 s. The gray vertical dashed lines are marked at 75 and 150 s for comparison. (**b**) The response at *t* = 75 s (reached atmospheric pressure; open symbols) and 150 s (closed symbols) from (**a**) as a function of the NO_2_ concentration.

**Figure 5 nanomaterials-13-00243-f005:**
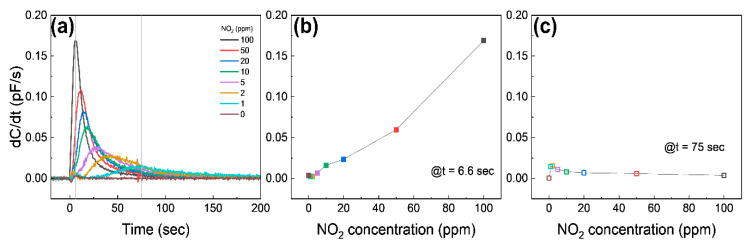
(**a**) Differential capacitance as a function of time at different NO_2_ concentrations. The gray solid lines are at 6.6 s and 75 s, respectively. (**b**,**c**) show the differential capacitance at 6.6 s and 75 s from (**a**), respectively.

**Figure 6 nanomaterials-13-00243-f006:**
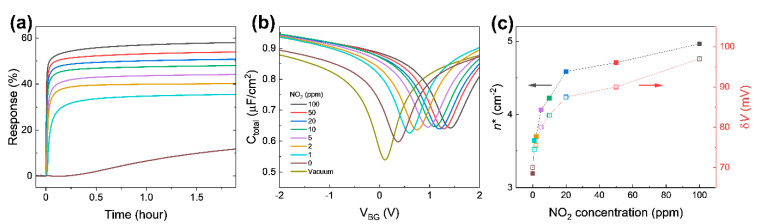
(**a**) Capacitive response at different NO_2_ concentrations for approximately 2 h. (**b**) *C*_total_ as a function of *V*_BG_ 2 h after exposure to different NO_2_ concentrations. (**c**) Extracted residual carrier density and potential variance from (**b**).

## Data Availability

Not applicable.

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
