# Peer review of "Capacitive NO2 Detection Using CVD Graphene-Based Device"

_nanomaterials, 2023, doi:10.3390/nano13020243_

Round 1
Reviewer 1 Report
The authors used the CVD to fabricate graphene and then showed the graphene-based FET for NO2 detection. The results demonstrate the feasibility of graphene-based FET sensor for NO2 gas, along with excellent linearity between capacitor and the substrate concentrations. This manuscript can be accepted after minjor revision because it is not well written and carefully edited with many grammar mistakes and confusing expressions. For example:
1. In the Abstract,
‘…various concentrations of O2 is also presented…’ should be ‘…various concentrations of NO2 are also presented…’
2. In the Keywords,
‘quantum capacitance’ should be ‘Quantum capacitance’
3. In the Introduction,
‘…a two-dimension carbon…’ should be ‘a two-dimensional carbon…’
4. Also in the Introduction,
‘Various gas molecules (NH3, H2O, CO, O2, NO2, etc.) can be adsorbed due to various
reasons on graphene surfaces’ should be ‘Various gas molecules (NH3, H2O, CO, O2, NO2, etc.) can be adsorbed on graphene surfaces due to various reasons’
5. In Section 3,
‘Once the graphene was exposed to 10-100 ppm NO2, dC/dt starts to immediately increase then reach the maximum, and then approach 0’
This sentence is Verb tense shifted.
6. Also in Section 3,
‘…induced by absorbed NO2…’ should be ‘…induced by adsorbed NO2…’
7. Figure 4 is missing in the manuscript.
8. some relevant papers about the capacitive properties of graphene-based device like Materials Today Sustainability, 2022, 17, 100096.; Electrochimica Acta 2015, 152, 216. can be cited in the main text.
Author Response
The authors thank Reviewer #1 for the insightful comments regarding submitted paper. Reviewer #1 also notes that the manuscript can be accepted with minor revision. Upon request, the authors made the relevant changes to the manuscript to improve the quality of our manuscript.
--------
Reviewer #1 mentioned a series of minor corrections listed below. All these points are corrected accordingly.
- In the Abstract,
‘…various concentrations of O2 is also presented…’ should be ‘…various concentrations of NO2 are also presented…’
- In the Keywords,
‘quantum capacitance’ should be ‘Quantum capacitance’
- In the Introduction,
‘…a two-dimension carbon…’ should be ‘a two-dimensional carbon…’
- Also in the Introduction,
‘Various gas molecules (NH3, H2O, CO, O2, NO2, etc.) can be adsorbed due to various
reasons on graphene surfaces’ should be ‘Various gas molecules (NH3, H2O, CO, O2, NO2, etc.) can be adsorbed on graphene surfaces due to various reasons’
- In Section 3,
‘Once the graphene was exposed to 10-100 ppm NO2, dC/dt starts to immediately increase then reach the maximum, and then approach 0’
This sentence is Verb tense shifted.
- Also in Section 3,
‘…induced by absorbed NO2…’ should be ‘…induced by adsorbed NO2…’
--------
Revewer #1 also mentioned a missing figure (Figure 4) in the manuscript.
Authors rechecked the original manuscript and found that the original manuscript contains the Figure 4 in its right place yet for somehow the distributed version has some missing information, such as missing Figure 4. We have included Figure 4 and its captions again in the revised version.
--------
Reviewer #1 recommended a couple of relevant references about the capacitive properties of graphene-based device like Materials Today Sustainability, 2022, 17, 100096.; Electrochimica Acta 2015, 152, 216.
We have included those references in our list:
They are listed as Reference #6 & Reference #7.
--------
Once again, we thank Reviewer #1 for constructive feedback on our manuscript.
Reviewer 2 Report
The manuscript is quite well prepared, however some additional enhancement and considerations are required:
1) The authors state that the gas sensor application of quantum capacitance effect of graphene-based devices is under investigated, which is quite true, but maybe the reference list can be improved.
Particularly the paper 10.1007/s10450-017-9895-0 perhaps may be included in the reference list and used for discussion of results.
2) The authors present the NO2 measurements in the nitrogen media. This is not very useful from the gas sensor application point of view as NO2 is usually needs to be measured in air. Some considerations in the influence of oxygen on the NO2 sensing are required.
3) NO2 is usually measured in quite low concentrations around 1 ppm. It would be great if authors could provide some considerations on the accuracy of measurements in this range.
Author Response
The authors thank Reviewer #2 for noting that our manuscript is well prepared. He/she mentioned some additional enhancement and considerations we need to consider in our manuscript.
--------
Reviewer #2 mentioned enhancing the list of references by adding more relevant reference, particularly the paper 10.1007/s10450-017-9895-0.
Authors found the proposed reference very useful. Authors are grateful to Reviewer #2 to mention the relevant reference so to enrich our list of references. Suggested paper is cited as Reference #11.
--------
Reviewer #2 raised a concern about NO2 related measurements in the nitrogen media since this condition may not be very useful from the gas sensor application point of view as NO2 usually needs to be measured in air. And so he/she requested some considerations on the influence of oxygen on the NO2 sensing are required.
This is our final goal for sure. Yet when it comes down to the adsorption process on the surface of graphene, graphene does not distinguish one type of molecules from another. So, as long as we are using pure graphene, it could only be ever long lasting question of which molecule is affecting the electronic state of graphene because there is no known way of distinguishing them from mesoscopic measurement results. We are planning the next step of improving our graphene-based device by chemically functionalizing its surface to increase the selectivity of adsorbed molecules on to the graphene surface while maintaining its electronic properties. But unfortunately, this is surely not within the scope of current manuscript.
However, we have added extra paragraph mentioning specific example related to this problem at the end of Introduction section (in page 2 lines 32 – 37).
--------
Reviewer #2 also mentioned the tested NO2 concentrations being rather high, and ask if lower dose NO2 data can be provided.
Authors also realized that the data points presented in our manuscript is all related to relatively larger concentration cases. Currently, we do not have the data with lower concentration case. However, we can make a careful deduction by comparing the linear trend of data with higher concentration of NO2 cases with 0 ppm case to safely state that the proposed device can measure the sub ppm concentration. We believe our deduction can be strengthened with the fact that the measured quantum capacitance effect is still within the distinguishable range for 0 ppm case. Related comment is added in our conclusion in page 8 lines 38 – 39.
In any ways, we would like to take this opportunity to thank Reviewer #2 for bringing up this point. We would definitely recalibrate the range of NO2 concentrations used for the future test of our improved devices.